# Postoperative avascular necrosis of the femoral head in pediatric femoral neck fractures

Yang Li, Dong Sun, Kelai Wang☉*, Jingwei Liu, Zhe Wang, Yu Liu

Department of Pediatric Surgery, Qilu Hospital of Shandong University, Shandong, China

* sddxqlyyxewkwkl@163.com

**Data Availability Statement:** The data that support the finding of this study are under the public moral protection.This data are available on reasonable request to the corresponding author(Email:

## Abstract

This study aimed to evaluate the relevant factors of postoperative avascular necrosis of the femoral head (AVN) in children with femoral neck fracture. This study retrospectively analyzed the clinical data of 28 children with femoral neck fractures treated at our center between July 2016 and January 2019. The average age was 9.3 (range, 4.4–14) years with 75% male participants. Fracture classification was based on the Delbet classification: there were four, seven, 15, and two cases of type I, II, III, and IV fractures, respectively. Displacement degree was based on the Garden classification. Sixteen cases had insignificant displacement (Garden types I and II), six had medium displacement (Garden type III), and six had significant displacement (Garden type IV). There were six early ($\leq$24 hours) and 22 delayed (>24 hours) surgeries. Twenty-three patients had satisfactory reduction, and five had unsatisfactory reduction. The mean postoperative follow-up period was 15.7 (range, 12–36) months. Follow-up was evaluated using the Ratliff scoring standards. The correlation between age, fracture classification, displacement degree, surgery timing, reduction quality, and other factors and AVN occurrence was statistically analyzed. Among 28 children, AVN was found in six cases. There were statistically significant differences in displacement degree (P = 0.001) and reduction quality (P = 0.001), while the occurrence of AVN did not significantly differ with sex (P = 0.117), age distribution (P = 0.218), fracture classification (P = 0.438), surgery timing (P = 0.255), and mechanism of injury (P = 0.436). The results of logistic regression analysis showed that displacement degree was a relevant risk factor (P = 0.049, odds ratio [OR] = 8.391, 95% confidence interval [CI]: 1.004−70.117), while reduction quality was not (P = 0.075, OR = 14.536, 95% CI: 0.757−278.928). Although the development of AVN in children with femoral neck fractures may be related to many factors, the results of this research suggest that there is a significant correlation between displacement degree and AVN occurrence.

## Introduction

Femoral neck fractures are rare in children, with an incidence of <1%. These fractures are mostly caused by high-energy injuries [1–7], such as fall injury, traffic accident trauma, and

sddxqlyyxewkwkl@163.com)and data administrator of Department of Imaging, Qilu Hospital of Shandong University(contact number:0531-82169305;Email: hzswk199312@163.com).The data are not publicly available due to the privacy information of patient, including name,contact details,home address and others.

**Funding:** The author(s) received no specific funding for this work.

**Competing interests:** The authors have declared that no competing interests exist.

crush by heavy objects. Due to the special anatomy of the proximal femur in children, the blood supply of the femoral head is susceptible to damage after fracture. Vascular injury to the femoral head is related to many kinds of complications, among which the most common and serious one is avascular necrosis of the femoral head (AVN) [1–7]. AVN in the pediatric population is a challenge without an ideal treatment strategy. Therefore, it is important to identify and reduce the risks of AVN after femoral neck fracture within the pediatric population. Many factors have been predicted in the development of AVN [1–5,7–21]. Among these factors, age, displacement degree, fracture classification, surgery timing, and reduction quality remain controversial. No authoritative and specific conclusions have been recognized widely. The main objective of this study was to identify the correlative factors of AVN after femoral neck fracture in children and to provide a reference for improving efficacy and prognosis. Through this study, we found that displacement degree is a correlative factor for AVN after femoral neck fracture in children.

## Materials and methods

### Study population

Medical records of children with femoral neck fractures admitted to our department between July 2016 and January 2019 were retrospectively reviewed. The inclusion criteria were as follows: (1) <18 years old; (2) surgery at Qilu Hospital of Shandong University; (3) injury caused by trauma; (4) closed fractures; and (5) postoperative follow-up time ≥12 months. The exclusion criteria were as follows: (1) open fractures; (2) pathological fractures; (3) no-surgery patients; (4) complicated with bone metabolic diseases. This research was approved by the Institutional Review Board of Qilu Hospital of Shandong University, and written informed consent was obtained from the parents of the patients.

### Fracture evaluation and treatment

Original, intraoperative, and postoperative radiographs were used to determine fracture classification, displacement degree, and reduction quality. Clinical data, imaging findings, and follow-up data were analyzed by three pediatric orthopedic surgeons who did not participate in the surgery, which excluded the interference of subjective factors. Discrepancies were resolved by discussion.

Fracture classification was based on the Delbet classification [6]—type I: epiphysiolysis fracture; type II: transfemoral neck fracture; type III: basilar femoral neck fracture; and type IV: intertrochanteric fracture.

Displacement degree was based on the Garden classification [22]—type I: incomplete fracture; type II: complete fracture but well-aligned without displacement; type III: partial displacement of the fracture, abduction of the femoral head, and mild external rotation and upward displacement of the femoral neck segment; and type IV: complete displacement of the fracture with significant external rotation and upward displacement of the femoral neck segment. Garden types I and II were classified as fractures with insignificant displacement. Garden type III was classified as fractures with medium displacement, and Garden type IV was classified as fractures with significant displacement. All participants were treated by closed surgery, which was adopted as the method of choice. Open surgery was the alternative in case of failure. All fractures were fixed by cannulated screws.

Reduction quality was evaluated by the femoral neck fracture reduction criteria proposed by Haidukewych et al. [23]—excellent: fracture displacement <2 mm or fracture angulation <5°; good: displacement 2–4 mm or angulation 5°–9°; moderate: displacement 5–10 mm or angulation 10°–20°; and poor: displacement >10 mm or angulation >20°. Patients with

**Table 1. Ratliff scoring standards.**

| Grade | Clinical manifestation | Imaging inspection |
|---|---|---|
| **Excellent** | No pain, Normal or slightly limited hip motion, Normal daily activity | Normal or mild deformity of the femoral neck |
| **Good** | Occasional pain, Limited hip joint movement <50%, Normal daily activities | Severe deformity of the femoral neck, mild osteonecrosis of the femoral head |
| **Poor** | Persistent pain, Limited hip movement >50%, Limited daily activities | Degenerative arthritis, joint fusion |

"excellent" fracture reduction quality were included in the satisfactory reduction group. Those with "good, moderate, and poor" reduction quality were deemed to have unsatisfactory reduction. All patients underwent hip "herringbone" plaster-assisted fixation. After healing of the fracture line, non-weight-bearing functional exercises were performed for at least one month. Thereafter, gradual weight-bearing functional exercises were performed regularly.

## Follow-up

All participants were followed up for a minimum period of 12 months, with an average follow-up duration of 15.6 (range,12–36) months. Follow-up was evaluated using the Ratliff criteria [24] (Table 1).

## Statistical analysis

Statistical analyses were performed using IBM SPSS Statistics for Windows, version 22.0 (IBM Corp., Armonk, N.Y., USA). Age, as a continuous characteristic, was compared using Mann–Whitney U test. Unordered categorical characteristics, including sex, mechanism of injury, fracture classification, surgery timing, and reduction quality, were compared using chi-squared test and Fisher's exact test. Displacement degree, as an ordinal categorical characteristic, was compared using Cochran-Armitage trend test. Displacement degree and reduction quality were further evaluated using logistic regression analysis. $P<0.05$ was considered statistically significant.

## Results

There were 28 patients who met the inclusion criteria, including 21 (75%) male and seven (25%) female patients. The median age was 9.4 (range, 4.4–14) years. The causes of the injuries were as follows: 23 (82.14%) cases of fall injury, three (10.71%) of traffic accident trauma, and two (7.14%) crushed by heavy objects. The average time between injury and surgery was 5.6 days, ranging from 12 hours to 25 days: six (21.43%) cases of early surgery (≤24 hours) and 22 (78.57%) of delayed surgery (>24 hours). Among the 28 participants, four (14.29%), seven (25%), 15 (53.57%), and two (7.14%) had types I, II, III, and IV fractures, respectively. Among the 28 participants, six (21.43%) had medium displacement, six (21.43%) had significant displacement, and 16 (57.14%) had insignificant displacement. A total of 26 patients underwent closed surgery successfully, and two underwent open surgery. There were 23 (82.14%) and five (17.86%) patients with satisfactory and unsatisfactory reductions, respectively.

The mean follow-up time for surgical outcome was 15.6 (range,12–36) months. According to the Ratliff scoring standards, at the last follow-up evaluation, 22, three, and three cases were

**Table 2. Analysis of factors related to avascular necrosis of the femoral head in 28 children.**

| Factor | Avascular necrosis of the femoral head | | Sum (N = 28) | P |
|---|---|---|---|---|
| | N (%) or Median (p25, p75) | | | |
| | No (N = 22) | Yes (N = 6) | | |
| **Age in years** | 7.5 (5.7, 12.1) | 11.05 (9.7, 12.5) | 9.4 (6.2, 12.1) | 0.218 |
| **Sex** | | | | |
| **Female** | 4 (18.18) | 3 (50) | 7 (25) | 0.117 |
| **Male** | 18 (81.82) | 3 (50) | 21 (75) | |
| **Displacement degree (Garden)** | | | | |
| **Insignificant displacement (Garden types I and II)** | 16 (72.73) | 0 (0) | 16 (57.14) | **0.001** |
| **Medium displacement (Garden type III)** | 4 (18.18) | 2 (33.33) | 6 (21.43) | |
| **Significant displacement (Garden type IV)** | 2 (9.09) | 4 (66.67) | 6 (21.43) | |
| **Fracture classification (Delbet)** | | | | |
| **I** | 2 (9.09) | 2 (33.33) | 4 (14.29) | 0.438 |
| **II** | 6 (27.28) | 1 (16.67) | 7 (25) | |
| **III** | 12 (54.54) | 3 (50) | 15 (53.57) | |
| **IV** | 2 (9.09) | 0 (0) | 2 (7.14) | |
| **Mechanism of injury** | | | | |
| **Fall injury** | 17 (77.27) | 6 (100) | 23 (82.14) | 0.436 |
| **Traffic accident trauma** | 3 (13.64) | 0 (0) | 3 (10.71) | |
| **Crush by heavy objects** | 2 (9.09) | 0 (0) | 2 (7.15) | |
| **Surgery timing** | | | | |
| **Early surgery** | 6 (27.27) | 0 (0) | 6 (21.43) | 0.255 |
| **Delayed surgery** | 16 (72.73) | 6 (100) | 22 (78.57) | |
| **Reduction quality** | | | | |
| **Satisfactory** | 21 (95.46) | 2 (33.33) | 23 (82.14) | **0.001** |
| **Unsatisfactory** | 1 (4.54) | 4 (66.67) | 5 (17.86) | |

P<0.05 was considered statistically significant.

excellent, good, and poor, respectively. The cumulative excellent and good rate was 89.29%. In the Mann–Whitney U test, chi-squared test, Fisher's exact test, and Cochran-Armitage trend test, there were significant differences in displacement degree (P = 0.001) and reduction quality (P = 0.001), while there were no significant differences in sex (P = 0.117), age (P = 0.218), fracture classification (P = 0.438), surgery timing (P = 0.255), and mechanism of injury (P = 0.436) between the two groups (Table 2).

Displacement degree and reduction quality were included in the logistic regression model to further identify correlative factors of osteonecrosis after femoral neck fracture. The analysis showed that displacement degree was a correlative risk factor for the development of osteonecrosis (P = 0.049, OR = 8.391, 95% CI: 1.004−70.117), whereas reduction quality was not (P = 0.075, OR = 14.536, 95% CI: 0.757−278.928) (Table 3).

**Table 3. Logistic regression analysis and proofreading of the degree of displacement and quality of reduction.**

| Risk factors | Odds ratio (OR) | 95% confidence interval (CI) | P |
|---|---|---|---|
| **Displacement degree** | 8.391 | 1.004−70.117 | **0.049** |
| **Reduction quality** | 14.536 | 0.757−278.928 | 0.075 |

P<0.05 was considered statistically significant.

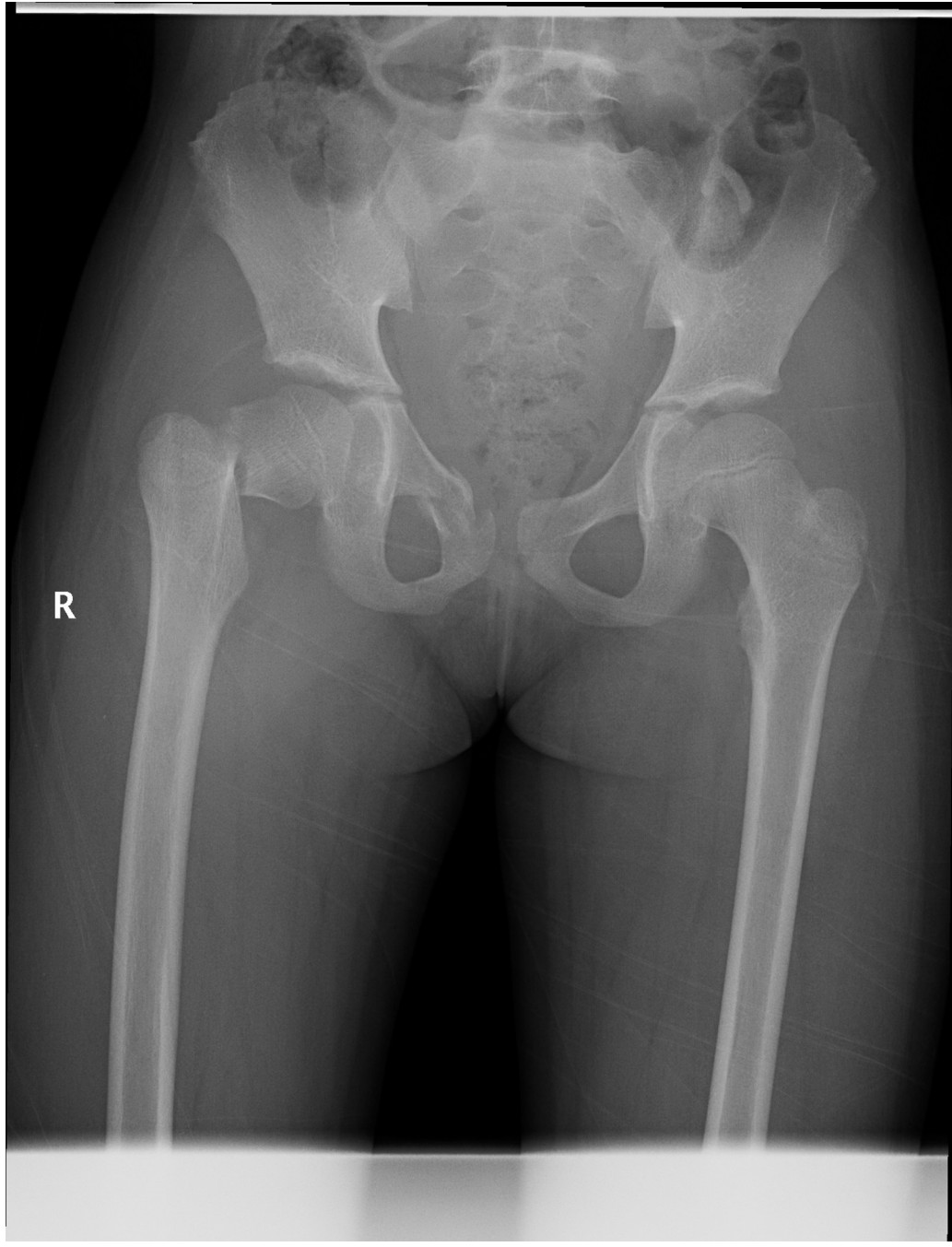

**Fig 1. Fracture classification and displacement degree.** Radiograph showing a Delbet type-III femoral neck fracture with significant displacement.

## Typical case 1

A 12-year-old girl was admitted to the hospital due to an accidental fall injury which occurred 2 days earlier. The patient had no pertinent medical history. Preoperative radiography revealed a Delbet type-III fracture with significant displacement. After admission, surgery was performed on post-injury day 2. Pre- and postoperative images are shown in Figs 1–3.

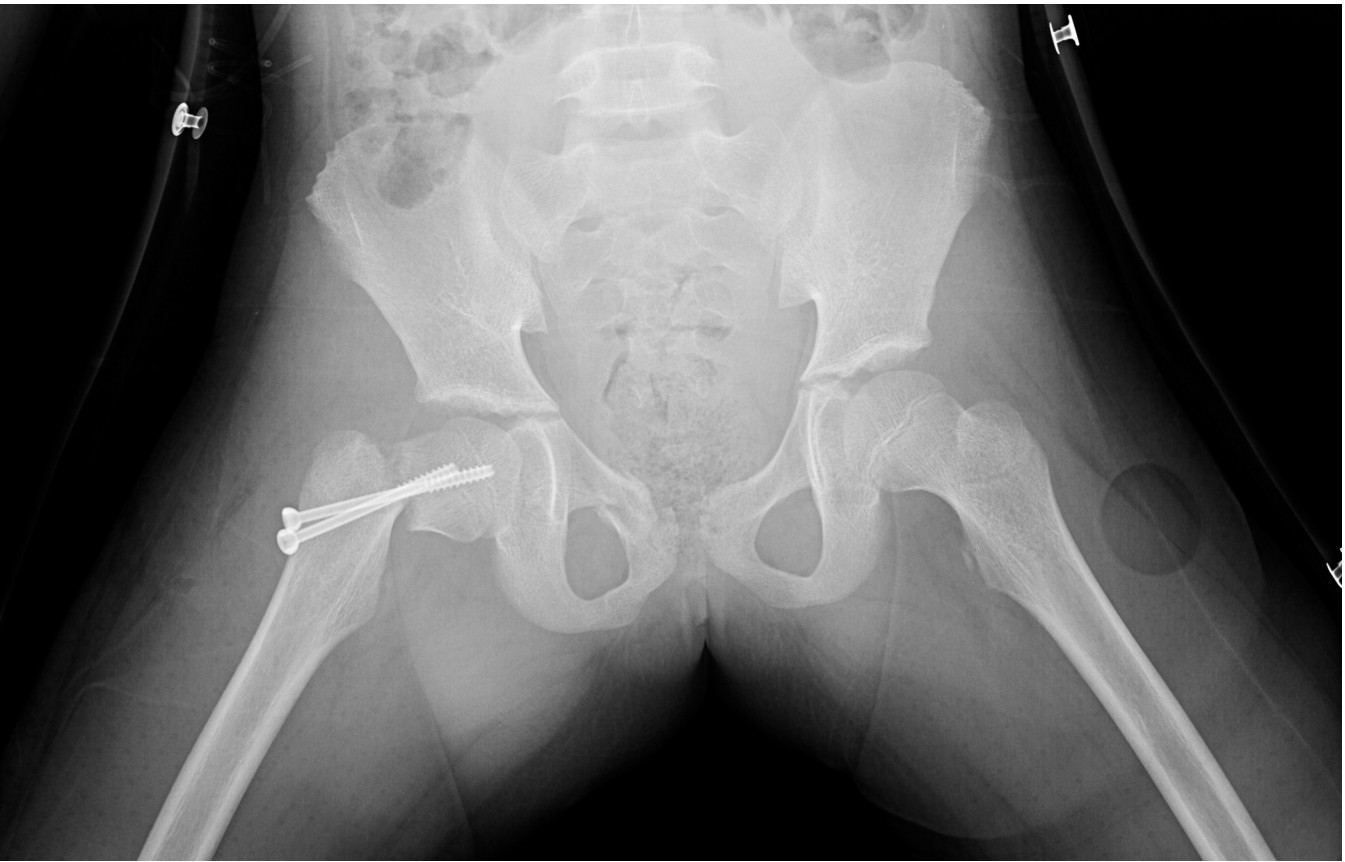

**Fig 2. Reduction quality.** Unsatisfactory reduction: The widest distance of the fracture ends is obvious (more than 10mm); the diaphragm is poorly aligned (more than 10˚).

### Typical case 2

A 11-year-old boy was admitted to the hospital due to an accidental fall injury which occurred eight hours earlier. The patient had no pertinent medical history. The preoperative radiography revealed a Delbet type-II fracture with insignificant displacement. After admission, relevant examinations were performed, and surgery was performed on post-injury 20 hours. Pre- and postoperative images are shown in Figs 4–6.

### Discussion

Pediatric femoral neck fractures are rare, comprising <1% of all pediatric fractures [1–7]. There are many opinions regarding surgery in these cases. Chen et al. [21] considered that open surgery was better than closed surgery, contrary to the results of a meta-analysis by Yerabosian et al. [4]. A meta-analysis by Alkhatib et al. [12] showed that there was no significant difference between the two surgical methods. In this study, all participants were treated using the same surgical protocol. Closed surgery was adopted as the method of choice. Open surgery was the alternative in case of failure. This ensured that no errors occurred due to variations in surgical method selection. Meanwhile, our literature review revealed several studies [25–27] comparing open and closed surgeries, which demonstrated a similar phenomenon; some cases with open surgeries had previously undergone closed surgeries. This research approach may be defective in theory, since the research on open or closed surgery should be completely

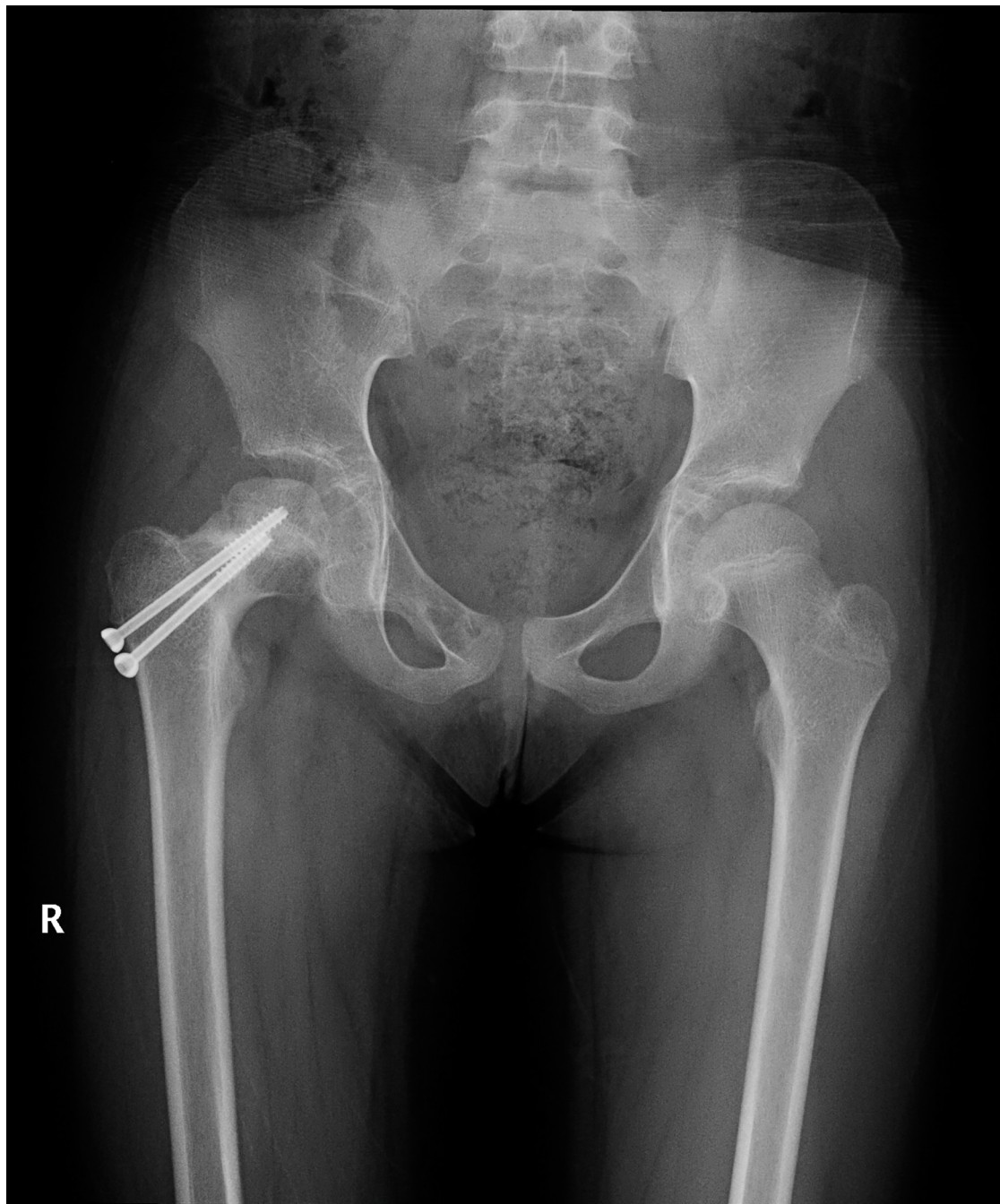

**Fig 3. The last follow-up evaluation at 12 months postoperatively.** Radiograph reveals irregular femoral head shapes, changes in density, and the presence of avascular necrosis of the femoral head.

independent and random. Setting three groups, including direct closed, direct open, and open surgery groups after closed failure, may be a better categorization. However, this would require a large sample size. There were only 28 cases in this study, and most parents refused direct open surgery. Therefore, we did not thoroughly study this aspect.

Regarding the implant, there is no evidence that different kinds of implants cause different effects on the postoperative complication rate [2,28]. Further, many researchers found that

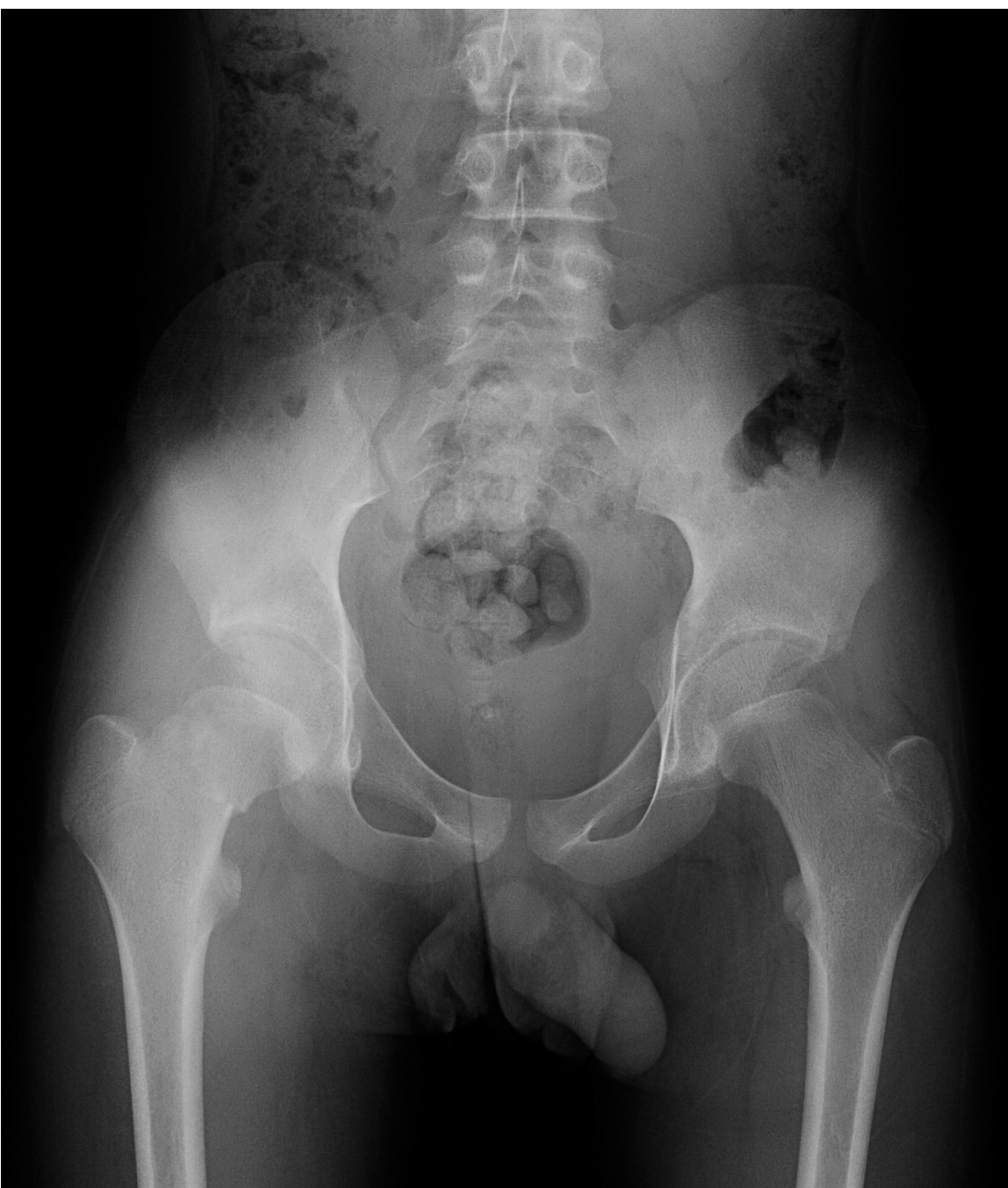

**Fig 4. Fracture classification and displacement degree.** Radiograph showing a Delbet type-II femoral neck fracture with insignificant displacement.

when the femoral neck fracture requires fixation across the epiphyseal plate, it is more important to ensure the stability of the fracture end than to protect the growth potential of the epiphyseal plate [5,29]. Therefore, all fractures in this study were fixed by cannulated screws, which are widely used.

In terms of postoperative protocol, few researchers have focused on the effect of early or late weight-bearing, while some studies have shown that the use of hip "herringbone" plaster

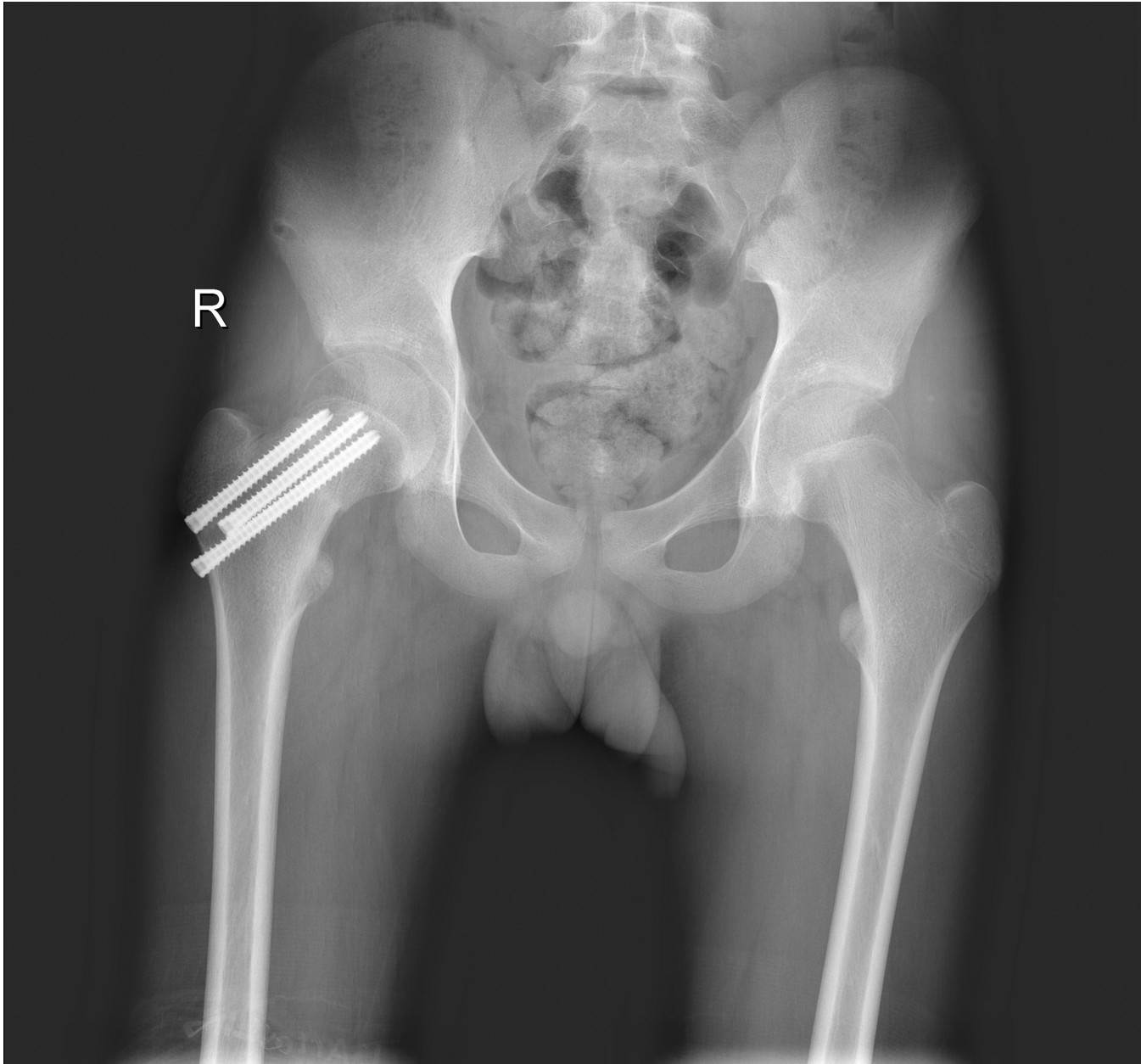

**Fig 5. Reduction quality.** Satisfactory reduction: The fractures ends have good alignment, no displacement and no angle.

could improve the results [16,28]; therefore, all participants whose records were evaluated in this study had undergone hip "herringbone" plaster-assisted fixation.

There are many complications related to femoral neck fracture in children. The most common and serious complication is AVN. Due to the particularity of femoral anatomy in children, the femoral head undergoes a series of changes throughout development; before three months of age, the inferior metaphyseal and lateral epiphyseal arteries are the main sources of blood to the developing femoral head. By the age of 18 months, the blood supply to the inferior metaphyseal artery begins to decrease, and the dominant supply shifts to the lateral epiphyseal vessels. By approximately four years of age, the lateral epiphyseal artery becomes the only

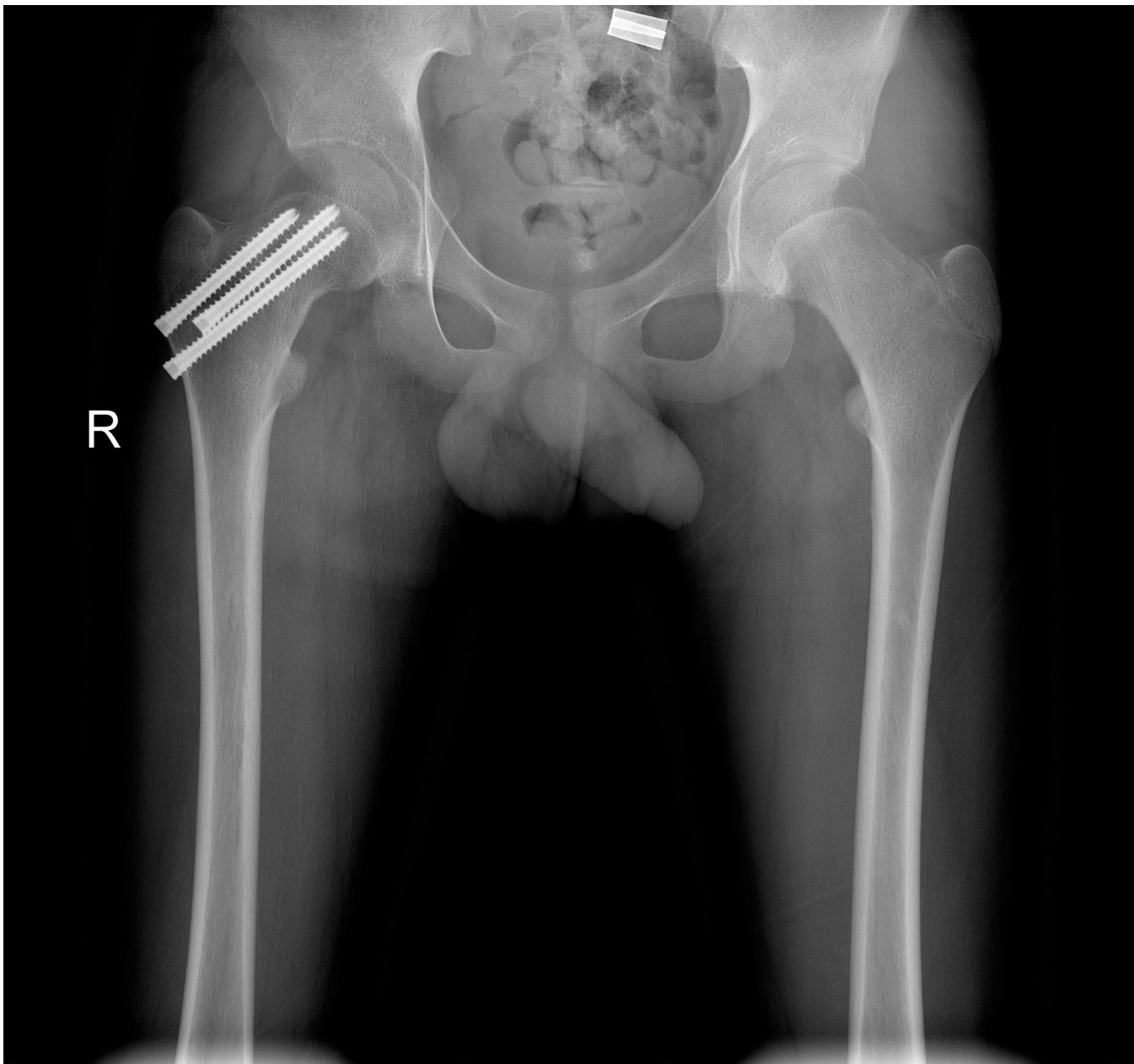

**Fig 6. The last follow-up evaluation at 12 months postoperatively.** Radiograph reveals regular femoral head morphology, uniform density, and no avascular necrosis of the femoral head.

source of blood to the femoral head. By the age of eight years, the round ligamentous artery of the femoral head, which would have developed physiological occlusion at the age of two years, begins to restore the blood supply to the femoral head. However, the supply is only to the area near the point of attachment of the round ligament. Until the epiphyseal plate heals during adolescence, the metaphysis remains blocked from the femoral head, making it extremely sensitive to vascular injury and prone to ischemic necrosis [8,9,30–32].

According to the literature, the incidence of AVN after femoral neck fracture in children varies from 17% to 90%, mainly occurring within one year after fracture [10,33–35]. Spence

et al. [2] found that the median duration of AVN occurrence after femoral neck fracture in children was 7.8 months. Therefore, the inclusion criteria of the present study required children to have been followed up for at least 12 months.

The occurrence of AVN is associated with multiple factors, among which age, displacement degree, fracture classification, surgery timing, and reduction quality remain controversial [1–5,7–21]. The present research mainly focused on these aspects.

## Age

Moon and Mehlman [3] conducted a meta-analysis based on 360 children with femoral neck fractures and found that the occurrence of AVN was closely related to age. The likelihood of AVN increased by 1.14 times for every one year of age. This may be due to the reconstructing and shaping ability of young children [3,7,12,36]. Some scholars consider that femoral head revascularization after femoral neck fracture is better in children <10 years of age [34,37]. Ratliff [24] reported that 29 of 30 children with AVN after femoral neck fracture were approximately 11 years of age. Wang et al. [13] concluded that the incidence of AVN increases with age, especially in patients of ≥12 years of age. Morsy [14] and Spence et al. [2] suggested that the occurrence of AVN is independent of age. Although there was no difference in the age distribution of AVN occurrence in our study, we found that the median age of children with AVN (11.1 years) was higher than that of children without AVN (7.5 years). All six children with AVN were of >10 years of age. Therefore, although the correlation between age and the occurrence of AVN needs to be further studied, high alertness for AVN should be maintained for children who suffer femoral neck fractures at older ages.

## Fracture classification (Delbet)

Six of the 28 children in this study had AVN, including two cases (50%) of type I, one (14.29%) of type II, and three (20%) of type III. This trend was similar to that in previous reports [15,37,38]. Delbet type-I fractures have a high risk of necrosis, ranging from 70% to 100% [15,37,38]. Therefore, vigilance is particularly needed in clinical practice. Additionally, the incidence of AVN is reported to be lower in types II and III fractures than in type-I fractures [3,10,34]. However, it is also noteworthy that type-III fractures tend to be located at the level of the basilar ring formed by the medial and lateral femoral circumflex arteries and are prone to vascular injury with AVN. Further, when the retinacular artery enters the articular capsule, it attaches to the bone and not the articular capsule. When a type-II femoral neck fracture occurs, the blood vessels are also easily damaged. Therefore, attention should also be paid to type-II and -III fractures. Type-IV fractures are extracapsular fractures that do not easily damage blood vessels directly and have a low incidence of AVN.

## Surgery timing

Most scholars currently consider that early treatment, especially within 24 hours [2,5,16–18] and as late as 48 hours [5,19], can reduce the incidence of AVN. Yeranosian et al. [4] analyzed the clinical data of 935 children and concluded that the incidence of AVN in delayed treatment (>24 hours) was 4.2 times than that in early treatment (≤24 hours). Bukva et al. [20] consider that the early operative time should be advanced to 12 hours. Alkhatib et al. [12] conducted a meta-analysis on the data of 231 children and found that AVN occurrence was not related to the time interval between injury and surgery. In this study, no AVN occurred in six children who underwent early surgery (≤24 hours). On the other hand, six children who underwent delayed surgery (>24 hours) had AVN. Statistical analysis in this study suggested that surgery timing was not related to AVN occurrence. However, it is difficult for some children to receive

early treatment (within 24 hours) due to remoteness of the accident scene, delay in referral to local hospitals, and priority of associated injury management. Therefore, the relationship between surgery timing and AVN occurrence still needs to be further studied by expanding the study sample capacity.

## Displacement degree

There is no internationally recognized evaluation standard for determining the degree of displacement of femoral neck fractures in children. The degree of fracture displacement in adults is mostly classified using the Garden classification [22]. Accordingly, Garden types I and II were classified as fractures with insignificant displacement, Garden type III was classified as fractures with medium displacement, and Garden type IV was classified as fractures with significant displacement. Quick et al. [11] concluded that the occurrence of AVN is associated with multiple factors. The most important factors are the degree of fracture displacement and degree of surrounding soft tissue injury at the time of the original injury. This may be due to more significant displacement directly damaging or distorting the major blood vessels supplying the femoral head, which leads to necrosis of the femoral head. Although several studies found displacement degree to be a key predictor in the development of AVN [2,13,14], other studies did not [3,34,36]. This lack of consensus may be due to several factors, such as inconsistent definitions of migration, variation in classification between observers, and small sample sizes. Our results revealed that necrosis occurred in two of six cases with medium displacement and four of six cases with significant displacement, while no necrosis occurred in the 16 cases with insignificant displacement. The results of the Cochran-Armitage trend test and logistic regression analysis showed statistical differences. This confirms that the displacement degree is a relevant risk factor for the development of AVN. Therefore, attention should be paid to the effect of the displacement degree on the prognosis of children with femoral neck fractures. Children with fractures with more severe displacement should be closely followed up for the presence of AVN.

## Reduction quality

Regardless of the fracture type, the best reduction quality is perfect anatomical reduction, although it is difficult to achieve this goal in practice. In other words, it is a kind of "ideal" state. Further, the concept of "anatomical reduction" lacks specific digital standards. In adults, the Garden alignment index is the most commonly used quality assessment method for femoral neck fracture reduction [39]. The Garden index is based on the inner edge of the femoral shaft and inner pressure bone trabecula of the medial femoral head. However, from birth, the neck-shaft angle in children is in the dynamic process of gradual decrease. Therefore, there are theoretical defects in judging the quality of reduction of femoral neck fractures in children using the Garden alignment index. There is no targeted reduction quality assessment method for femoral neck fractures in children, and most scholars choose the evaluation criteria that Haidukewych et al. [23] proposed [2,10]. In this study, the Fisher exact test showed a statistically significant difference between the incidence of necrosis in the satisfactory (8.7%) and unsatisfactory (80%) reduction groups. This implied that reduction quality may be a relevant factor for AVN, and better reduction quality may reduce the risk of AVN. These findings are similar to those previously reported [10,14]. However, in logistic regression analysis, reduction quality was not a correlative risk factor for the development of osteonecrosis. These results are also similar to those of previous studies [2,3]. The small sample size may be an important reason for this phenomenon. Besides, in clinical practice, satisfactory reduction is more likely to be achieved in patients with insignificant displacement, while unsatisfactory reduction often

occurs in the group of medium and significant displacement cases. There may be a potential relevance between these two factors, which causes the difference between the results of Fisher's exact test and logistic regression analysis. Despite this, attention still needs to be paid to reduction quality. Satisfactory reduction should be achieved as much as possible to possibly reduce the occurrence of AVN.

There are some limitations to this study. Insufficient sample size and uneven distribution could have caused a reduction of the statistical power. Additionally, this was a retrospective study, and the follow-up time of some patients was short, which has a certain effect on the accuracy of study results. However, due to the rarity of this fracture, these limitations may exist in the study of any single institution. Multi-center prospective research can address these limitations and provide better and clearer answers.

## Conclusions

Our study results suggest that displacement degree is a relevant risk factor for the development of osteonecrosis. Fractures with more severe displacement are more likely to develop AVN. However, satisfactory reduction quality may influence the occurrence of AVN. The correlation between age, fracture classification, surgery timing, and other factors and AVN needs to be assessed further in future studies.

## Acknowledgments

I would like to thank all my teachers in Qilu Hospital of Shandong University who have helped me in developing the fundamental and essential academic competence. I would also like to thank the staff of the imaging department and medical records room in Qilu Hospital of Shandong University, who provided the specific data. Finally, I appreciate the study participants for allowing me to use their data; without their cooperation, we could not have completed this study.

## Author Contributions

**Conceptualization:** Dong Sun.

**Data curation:** Yang Li.

**Formal analysis:** Yang Li.

**Investigation:** Zhe Wang.

**Methodology:** Jingwei Liu.

**Project administration:** Dong Sun, Kelai Wang.

**Resources:** Yu Liu.

**Software:** Yang Li.

**Supervision:** Kelai Wang.

**Validation:** Kelai Wang.

**Visualization:** Jingwei Liu.

**Writing – original draft:** Dong Sun.

**Writing – review & editing:** Yang Li.

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
