## [Decision Letter · Decision Letter 0]

25 Jan 2021

PONE-D-21-00008

Postoperative avascular necrosis of the femoral head in pediatric femoral neck fractures

PLOS ONE

Dear Dr. Wang,

Thank you for submitting your manuscript to PLOS ONE. After careful consideration, we feel that it has merit but does not fully meet PLOS ONE’s publication criteria as it currently stands. Therefore, we invite you to submit a revised version of the manuscript that addresses the points raised during the review process.

We look forward to receiving your revised manuscript.

Kind regards,

Wen-Jun Tu

Academic Editor

PLOS ONE

Journal Requirements:

Reviewers' comments:

Reviewer's Responses to Questions

**Comments to the Author**

1. Is the manuscript technically sound, and do the data support the conclusions?

Reviewer #1: Partly

2. Has the statistical analysis been performed appropriately and rigorously? 

Reviewer #1: Yes

3. Have the authors made all data underlying the findings in their manuscript fully available?

Reviewer #1: Yes

4. Is the manuscript presented in an intelligible fashion and written in standard English?

Reviewer #1: Yes

5. Review Comments to the Author

Reviewer #1: This paper describes the high risk of developing AVN with more displaced femoral neck fractures (type III and IV) and no incidences in minimally displaced fractures (type I & II). The sample size is quite small compared to other studies that were referenced (28 total with 6 patients having AVN). When looking at reduction quality as an independent factor it would help to determine that in the setting of the degree of displacement as that will likely be directly correlative (ie. less displacement better reduction) therefore this should be included in the analysis - this appears to be addressed by table 3 but it is not clear to the reader. The table formatting is very difficult to follow due to it's embedding within the manuscript - maybe consider changing to a box plot figure demonstrating the correlation coefficient. There are multiple places within the manuscript where words are not spaced apart (ie. page 6 line 112; page 12 line 230). On page 3 line 45 would change this to read "vascular injury to the femoral head is related...

you do reasonably acknowledge the limitations of the study and mention the option of a multicenter study to increase the statistical power.

6. PLOS authors have the option to publish the peer review history of their article (what does this mean?). If published, this will include your full peer review and any attached files.

Reviewer #1: No

---

## [Author Response · Author response to Decision Letter 0]

31 Dec 2021

[Date of Resubmission]

Emily Chenette

Deputy Editor-in-Chief

PLOS ONE

Dear Wen-Jun Tu,

Thank you for your letter and for the reviewers’ comments concerning our manuscript entitled "Postoperative avascular necrosis of the femoral head in pediatric femoral neck fractures" (ID: PONE-D-21-00008).

We have studied the reviewers' comments carefully and have revised our manuscript accordingly, which we hope will meet with approval. Revised texts are marked in red in the manuscript. The main revisions in the manuscript and our responses to the reviewers’ comments are provided below.

Once again, thank you for the letter and for your time.

Sincerely,

Dr. Kelai Wang

Department of Pediatric Surgery, Qilu Hospital of Shandong University

107 Wenhua West Road, Lixia District, Jinan, Shandong 250012, China

Telephone: 86-18560082255

Fax: none

Email: sddxqlyyxewkwkl@163.com

1. 

Response: Thank you for your comment. We have ensured that our manuscript meets the style requirement of PLOS ONE.

Response: Thank you for your comment. Because these data contain some sensitive information of patients, such as name, home address, career, and others, they are only stored in the Department of Imaging, Qilu Hospital of Shandong University. These data are available upon reasonable request from the Ethics Committee on Scientific Research of Shandong University Qilu Hospital. If the readers need the data of this study, they should contact the host of this project, Dr. Wang (Email: sddxqlyyxewkwkl@163.com). 

Reviewers' comments:

Reviewer's Responses to Questions

Comments to the Author

1. Is the manuscript technically sound, and do the data support the conclusions?

Reviewer #1: Partly

Response: Thank you for your suggestion. We believe that despite less satisfactory statistical efficiency due to the small number of cases, our data analysis of existing cases is rigorous and credible. Therefore, the result of this study is credible.

2. Has the statistical analysis been performed appropriately and rigorously?

Reviewer #1: Yes

Response: Thank you for your kind comment.

3. Have the authors made all data underlying the findings in their manuscript fully available?

Reviewer #1: Yes

Response: Thank you for your kind comments.

4. Is the manuscript presented in an intelligible fashion and written in standard English?

Reviewer #1: Yes

Response: Thank you for your kind comment.

5. Review Comments to the Author

Reviewer #1: This paper describes the high risk of developing AVN with more displaced femoral neck fractures (type III and IV) and no incidences in minimally displaced fractures (type I & II). The sample size is quite small compared to other studies that were referenced (28 total with 6 patients having AVN). When looking at reduction quality as an independent factor it would help to determine that in the setting of the degree of displacement as that will likely be directly correlative (ie. less displacement better reduction) therefore this should be included in the analysis - this appears to be addressed by table 3 but it is not clear to the reader. The table formatting is very difficult to follow due to it's embedding within the manuscript - maybe consider changing to a box plot figure demonstrating the correlation coefficient. There are multiple places within the manuscript where words are not spaced apart (ie. page 6 line 112; page 12 line 230). On page 3 line 45 would change this to read "vascular injury to the femoral head is related...

you do reasonably acknowledge the limitations of the study and mention the option of a multicenter study to increase the statistical power.

Response:

Thank you for reviewing our manuscript, ID# PONE-D-21-00008, from which we have benefited both academically and logically. We apologize for the late reply. The major reason for this is that instead of the originally scheduled date in March this year, the submission date of this manuscript was postponed twice due to some reasons. Concerning the issues you pointed and the ones we discovered during revision, our responses are provided below.

Once again, thank you for your constructive comments and helpful suggestions.

First, in terms of the sample size, the low incidence of this disease and widespread global pandemic have made it difficult to collect related cases, which is one of the reasons we repeatedly postponed submitting this manuscript. Therefore, it is quite challenging to expand the incidence size in the short term. 

Through an intensive review of related literature, we found that because of the low incidence of this disease, several single-center studies on femoral neck fractures also encountered such difficulty of limited sample size. Some of these articles include: 

Article 1: Dai ZZ, Zhang ZQ, Ding J, Wu ZK, Yang X, Zhang ZM, et al. Analysis of risk factors for complications after femoral neck fracture in pediatric patients. J Orthop Surg Res. 2020;15(1): 58. doi: 10.1186/s13018-020-01587-9. PMID: 32075662; PMCID: PMC7029480. 

The sample size of this article was 44 cases. 

Article 2: Eamsobhana P, Keawpornsawan K. Nonunion paediatric femoral neck fracture treatment without open reduction. Hip Int. 2016;26(6): 608-611. doi: 10.5301/hipint.5000382. Epub 2016 May 23. PMID: 27229168. 

Nine cases were included in this article. 

Article 3: Bukva B, Abramović D, Vrgoč G, Marinović M, Bakota B, Dučić S, et al. Femoral neck fractures in children and the role of early hip decompression in final outcome. Injury. 2015;46 Suppl 6: S44-47. doi: 10.1016/j.injury.2015.10.059. Epub 2015 Nov 17. PMID: 26592094. 

Twenty-eight cases were included in this article.

As the project proceeds, related cases are still being included in this research. Nevertheless, because the follow-up time of some new cases does not meet the inclusion criteria, the sample size in this study has not yet been increased. In the future, more cases are certain to be included to expand the sample size, based on which we will conduct another round of statistical analysis to arrive at more accurate and representative conclusions. 

Second, concerning the inappropriateness in Form 3, we consulted some professionals in statistics and made careful revisions thereafter. In addition, we further classified the degree of fracture displacement in detail according to the Garden fracture classification: Garden types I and II were regarded as insignificant displacement fractures, Garden type III was regarded as a medium displacement fracture, and Garden type IV was regarded as significant displacement fractures. This led the results of the Fisher exact test and multivariate regression analysis to be more accurate and rigorous. Furthermore, we found that the degree of displacement is an independent risk factor for avascular necrosis of the femoral head, and the quality of reduction is only statistically meaningful in the Fisher exact test, which implies that the main factor for avascular necrosis of the femoral head is fracture displacement and the impact of the reduction quality is relatively small. 

Accordingly, we have revised the related parts in the manuscript. 

Meanwhile, we noticed that the 95% confidence interval after binary log analysis is 1.004−70.117. Initially, we thought this high value would affect our conclusions. However, the reference and consultation of similar research literature showed that this may be due to the limited number of cases and the low incidence of such disease. Therefore, such a phenomenon is common in the statistical operations of similar studies: 

Article 1: (Spence D, DiMauro Jp, Miller PE, Glotzbecker MP, Hedequist DJ, Shore BJ. Osteonecrosis after femoral neck fractures in children and adolescents: analysis of risk factors. J Pediatr Orthop. 2016;36(2): 111-116. doi: 10.1097/BPO.0000000000000424. PMID: 25730381.) 

The 95% confidence interval in this article is similar to that of the present study; all approximately 1−70, some even reached 1.08−175.67.

Article 2: (Moon ES, Mehlman CT. Risk factors for avascular necrosis after femoral neck fractures in children: 25 Cincinnati cases and meta-analysis of 360 cases. J Orthop Trauma. 2006;20(5): 323-329. doi: 10.1097/00005131-200605000-00005. PMID: 16766935.)

The 95% confidence interval in this article is also relatively large, ranging from 3.53 to 59.85.

Article 3: (Li Z, Zhuang Z, Hong Z, Chen L, He W, Wei Q. Avascular necrosis after femoral neck fracture in children and adolescents: poor prognosis and risk factors. Int Orthop. 2021;45(11): 2899-2907. doi: 10.1007/s00264-021-05210-2. Epub 2021 Sep 21. PMID: 34549321.) 

The 95% confidence interval in this article is also relatively large, which is 2.31−129.56.

Article 4: (Yeranosian M, Horneff JG, Baldwin K, Hosalkar HS. Factors affecting the outcome of fractures of the femoral neck in children and adolescents: a systematic review. Bone Joint J. 2013;95-B[1]: 135-142. doi: 10.1302/0301-620X.95B1.30161. PMID: 23307688.) 

This article summarizes many previous studies and discovers that the calculated 95% confidence interval values of risk factors have a large range.

Therefore, we believe that despite less satisfactory statistical efficiency due to the small number of cases, our data analysis of existing cases is similar to that in the above-mentioned literature, which is also rigorous and credible. 

Third, we have revised the manuscript accordingly. The revised texts are marked in red. Thank you for carefully reviewing our manuscript. 

Fourth, we adjusted the structure of the article in line with the general structure of academic papers.

The above are my responses to the questions raised by the expert reviewers. We would like to thank the reviewers and editors again for their valuable comments and suggestions, which were valuable in improving the quality of our manuscript.

---

## [Decision Letter · Decision Letter 1]

21 Feb 2022

PONE-D-21-00008R1Postoperative avascular necrosis of the femoral head in pediatric femoral neck fracturesPLOS ONE

Dear Dr. Wang,

Thank you for submitting your manuscript to PLOS ONE. After careful consideration, we feel that it has merit but does not fully meet PLOS ONE’s publication criteria as it currently stands. Therefore, we invite you to submit a revised version of the manuscript that addresses the points raised during the review process.

We look forward to receiving your revised manuscript.

Kind regards,

Wen-Jun Tu

Academic Editor

PLOS ONE

Reviewers' comments:

Reviewer's Responses to Questions

**Comments to the Author**

1. If the authors have adequately addressed your comments raised in a previous round of review and you feel that this manuscript is now acceptable for publication, you may indicate that here to bypass the “Comments to the Author” section, enter your conflict of interest statement in the “Confidential to Editor” section, and submit your "Accept" recommendation.

Reviewer #1: All comments have been addressed

2. Is the manuscript technically sound, and do the data support the conclusions?

Reviewer #1: Partly

3. Has the statistical analysis been performed appropriately and rigorously? 

Reviewer #1: I Don't Know

4. Have the authors made all data underlying the findings in their manuscript fully available?

Reviewer #1: Yes

5. Is the manuscript presented in an intelligible fashion and written in standard English?

Reviewer #1: Yes

6. Review Comments to the Author

Reviewer #1: Overall, the manuscript reads better with the corrections. I still have several concerns however.

Throughout the manuscript there are issues with spacing and punctuation (no space between a word and the parentheses), spaces before commas, etc. This needs to be edited more carefully.

Abstract: Check spaces throughout the result section. Given that the number of children who has AVN was 6, you cannot state that this study proved correlation between degree of displacement and AVN, you can state the results suggest a correlation and there was significance. The sample size is far to small to claim proof of independent correlation.

methods: p6 line 103: should state closed reduction and open reduction with or without fixation.

results: p8 line 137-138: eliminate trailing 0s (75% instead of 75.000%), results with non-round numbers should be 2 digits beyond the decimal (7.14% instead of 7.143%). Line 140: "bruising by heavy objects" unlikely to cause a femoral fracture, was it that the object fell on top of the patient? Please clarify this. page9 Line 159 should be correlative factors.

page 12: figure legends should be all together at the end of the manuscript after the references and before the actual figures. Discussion: good overview of influence of growth on the femoral neck, explanation of the classification schema used and review of existing literature.

figures: good demonstration of fracture displacement

tables: should all go at the end of the manuscript. The journal editors usually place the tables within the manuscript appropriately.

7. PLOS authors have the option to publish the peer review history of their article (what does this mean?). If published, this will include your full peer review and any attached files.

Reviewer #1: **Yes: **Eileen Raynor

---

## [Author Response · Author response to Decision Letter 1]

3 Apr 2022

[Date of submission:2022/4/5]

Emily Chenette

Editor-in-Chief

PLOS ONE

Dear Editor

Thank you for your letter and for the reviewer‘s comments concerning our manuscript entitled“Postoperative avascular necrosis of the femoral head in pediatric femoral neck fractures (ID:PONE-D-21-00008R1). Those comments are all valuable and very helpful for revising and improving our paper, as well as the important guiding significance to our researches. We have studied comments carefully and have made correction which we hope meet with approval. Revised portion are marked in red in the paper. The main corrections in the paper and the responds to the reviewer’s comments are as flowing:

Responds to the reviewer‘s comments:

Reviewer #1:

1.Throughout the manuscript there are issues with spacing and punctuation (no space between a word and the parentheses), spaces before commas, etc. This needs to be edited more carefully

Response: Thank you for the suggested. The Manuscript had been edited carefully to move this manuscript closer to publication in the PLOS ONE

2: Abstract: Check spaces throughout the result section. Given that the number of children who has AVN was 6, you cannot state that this study proved correlation between degree of displacement and AVN, you can state the results suggest a correlation and there was significance. The sample size is far to small to claim proof of independent correlation.

Response: We are grateful for the suggestion. To be more clear and preciseness, We revised the part of Abstract :”Although the development of AVN in children with femoral neck fractures may be related to many factors, the results of this research suggest that there is a significant correlation between displacement degree and AVN occurrence

3: methods: p6 line 103: should state closed reduction and open reduction with or without fixation. 

results: p8 line 137-138: eliminate trailing 0s (75% instead of 75.000%), results with non-round numbers should be 2 digits beyond the decimal (7.14% instead of 7.143%). page9 Line 159 should be correlative factors.

Response：Special thanks to you for your good comments. We have re-written this part according to the Reviewer’s suggestion

4: Line 140: "bruising by heavy objects" unlikely to cause a femoral fracture, was it that the object fell on top of the patient? Please clarify this. 

Response： I'm sorry for what happened. Because of my confusion, I should have written "crushed by heavy objects "instead of "bruising by heavy objects". It has now been modified. We appreciate for your warm work earnestly, and hope that the correction will meet with approval.

The manuscript has been rechecked and the necessary changes have been made in accordance with the reviewer’s suggestions. Thank you for your consideration. I look forward to hearing from you.

Author’s name: Kelai Wang

Affiliation: Department of Pediatric Surgery, Qilu Hospital of Shandong University, 107 Wenhuaxi Road, Jinan, Shandong, China

Postal address:250012

Phone number: +8618560082255

Email address:sddxqlyyxewkwkl@163.com

---

## [Decision Letter · Decision Letter 2]

22 Apr 2022

Postoperative avascular necrosis of the femoral head in pediatric femoral neck fractures

PONE-D-21-00008R2

Dear Dr. Wang,

We’re pleased to inform you that your manuscript has been judged scientifically suitable for publication and will be formally accepted for publication once it meets all outstanding technical requirements.

Kind regards,

Wen-Jun Tu

Academic Editor

PLOS ONE

Additional Editor Comments (optional):

Reviewers' comments:

Reviewer's Responses to Questions

**Comments to the Author**

1. If the authors have adequately addressed your comments raised in a previous round of review and you feel that this manuscript is now acceptable for publication, you may indicate that here to bypass the “Comments to the Author” section, enter your conflict of interest statement in the “Confidential to Editor” section, and submit your "Accept" recommendation.

Reviewer #1: (No Response)

2. Is the manuscript technically sound, and do the data support the conclusions?

Reviewer #1: Yes

3. Has the statistical analysis been performed appropriately and rigorously? 

Reviewer #1: Yes

4. Have the authors made all data underlying the findings in their manuscript fully available?

Reviewer #1: Yes

5. Is the manuscript presented in an intelligible fashion and written in standard English?

Reviewer #1: Yes

6. Review Comments to the Author

Reviewer #1: thank you for your attention to the revisions, it now reads much better. The only correction is in Table 2 where mechanism of injury still reads "bruising by heavy objects". please correct that in the table to match crush injury as in the results section.

7. PLOS authors have the option to publish the peer review history of their article (what does this mean?). If published, this will include your full peer review and any attached files.

Reviewer #1: No

---

## [Editor Report · Acceptance letter]

4 May 2022

PONE-D-21-00008R2 

Postoperative avascular necrosis of the femoral head in pediatric femoral neck fractures 

Dear Dr. Wang:

I'm pleased to inform you that your manuscript has been deemed suitable for publication in PLOS ONE. Congratulations! Your manuscript is now with our production department. 

Kind regards, 

on behalf of

Dr. Wen-Jun Tu 

Academic Editor

PLOS ONE